# Polyacetylenes from the Roots of *Swietenia macrophylla* King

**DOI:** 10.3390/molecules24071291

**Published:** 2019-04-02

**Authors:** Cheng-Neng Mi, Hao Wang, Hui-Qin Chen, Cai-Hong Cai, Shao-Peng Li, Wen-Li Mei, Hao-Fu Dai

**Affiliations:** 1Key Laboratory of Biology and Genetic Resources of Tropical Crops, Ministry of Agriculture, Institute of Tropical Bioscience and Biotechnology, Chinese Academy of Tropical Agricultural Sciences, Haikou 571101, China; michengnengnpc@126.com (C.-N.M.); wanghao@itbb.org.cn (H.W.); chenhuiqin@itbb.org.cn (H.-Q.C.); caicaihong@itbb.org.cn (C.-H.C.); 2Institute of Tropical Agriculture and Forestry, Hainan University, Haikou 570228, China; lisp555@126.com

**Keywords:** *Swietenia macrophylla*, roots, polyacetylene, cytotoxicity

## Abstract

A phytochemical investigation of the roots of *Swietenia macrophylla* led to the isolation of seven polyacetylenes, including five new compounds (**1**–**5**) and two known ones (**6**–**7**). Their structures were elucidated by extensive spectroscopic analysis and detailed comparison with reported data. All the isolates were tested for their cytotoxicity against the human hepatocellular carcinoma cell line BEL-7402, human myeloid leukemia cell line K562, and human gastric carcinoma cell line SGC-7901. Compounds **1** and **6** showed moderate cytotoxicity against the above three human cancer cell lines with IC_50_ values ranging from 14.3 to 45.4 μM. Compound **4** displayed cytotoxicity against the K562 and SGC-7901 cancer cell lines with IC_50_ values of 26.2 ± 0.4 and 21.9 ± 0.3 μM, respectively.

## 1. Introduction

Natural polyacetylenes, structures featuring two or more triple bonds [1], are mainly found in plants belonging to the Araliaceae [2,3,4,5,6,7], Asteraceae [8,9,10], Umbelliferae [11,12,13], Santalaceae [14], Pittosporaceae [15], and Oleaceae [16] families. However, polyacetylenes are uncommon in the Meliaceae family, with only six such compounds having been found in total in this family, one of which is from *Swietenia mahagoni* [17], three from *Toona ciliate* [18], and two from the stem bark of *Khaya ivorensis* A. Chev [19,20]. Naturally occurring polyynes are classified into four types: acyclic C_18_–C_14_ acetylene compounds; acyclic C_13_–C_8_ acetylene compounds; compounds with an allene substructure; and aromatic and heterocyclic acetylene compounds [1]. Diverse polyacetylene structures exhibit a series of bioactivities, including cytotoxicity [8,10,11,21] and antimicrobial [14,22,23], antiviral [24], and enzyme-inhibitory [6,25] activities.

*Swietenia macrophylla*, a perennial deciduous timber tree that reaches a height of up to 50 m [26], is native to Central and South America [27] and widely distributed in West India, Malaysia, and southern China [28,29]. Antecedent chemical investigations on *S. macrophylla* have focused mostly on the aboveground parts and their bioactive limonoids [30]. It is necessary to expand the scope of research on *S. macrophylla* and discover or develop additional biologically active constituents of this plant genus [31]. Our recent study on the roots of *S. macrophylla* led to the isolation of a series of xanthones, limonoids, and other chemical components [32,33]. As a continuation of our studies on the biologically active agents from this plant, five new and two known acyclic C_18_–C_14_ polyacetylenes have been further isolated here, and their cytotoxic activities against the human hepatocellular carcinoma cell line BEL-7402, human myeloid leukemia cell line K562, and human gastric carcinoma cell line SGC-7901 were investigated. In this paper, the isolation, structural elucidation, and cytotoxicity of these compounds are reported as follows.

## 2. Results and Discussion

The chemical examination of the ethyl acetate (EtOAc) extract from the roots of *S. macrophylla* resulted in the isolation and identification of five new polyacetylenes (Figure 1), respectively named heptadeca-9-ene-4,6-diyne-3,11-diol (**1**), (*E*)-heptadeca-8-ene-4,6-diyne-3,10,11-triol (**2**), 10-methoxyheptadeca-4,6-diyne-3,9-diol (**3**), tetradeca-1,3-diyne-6,7,8-triol (**4**) and 6,7,8,9-tetraacetoxytetradeca-1,3-diyne (**5**), together with two known compounds, which were identified as *α*-hexy-3-(6-hydroxy-2,4-ocadiynyl)oxiranemethanol (**6**) [17] and (3*R*,8*E*,10*S*)-heptadec-8-ene-4,6-diyne-3,10-diol (**7**) [18] by comparing their experimental spectroscopic data with the reported data in the literature. HRESIMS and NMR spectra for compounds **1**–**5** are shown in the Appendix A.

Compound **1** was obtained as a yellow oil with a positive optical rotation
${\left[\alpha \right]}_{D}^{25}$ +10 (*c* 0.1, CH_3_OH) and has a molecular formula of C_17_H_26_O_2_, as evidenced by the HRESIMS peak at *m*/*z* 285.1830 [M + Na]^+^ (calcd 285.1831 for C_17_H_26_NaO_2_), requiring five unsaturation degrees. The UV spectrum displayed typical absorption bands for a conjugated ene-yne-yne chromophore at λ_max_ 280 and 270 nm [11], and the IR spectrum showed the OH group (3435 cm^−1^), triple-bond (2234 cm^−1^), and olefinic double-bond (1639 cm^−1^) absorptions. The ^1^H-NMR data (Table 1) showed the presence of two olefinic protons at δ_H_ 5.50 (1H, m, overlapped, H-9) and 5.48 (1H, m, overlapped, H-10); two oxymethine protons, which appeared at δ_H_ 4.34 (1H, t, *J* = 6.4 Hz, H-3) and 4.39 (1H, q, *J* = 6.7 Hz, H-11); two triplet methyl groups at δ_H_ 0.99 (3H, t, *J* = 7.4 Hz, H-1) and 0.87 (3H, t, *J* = 6.8 Hz, H-17); and fourteen aliphatic methylene protons at δ_H_ 1.72 (2H, m, H-2), 3.09 (2H, dd, *J* = 5.0, 10.2 Hz, H-8), 1.60 (1H, m, H-12a), 1.44 (1H, m, H-12b), and 1.27−1.28 (8H, m, overlapped, H-13−16). A detailed analysis of the ^13^C-NMR and DEPT spectra (Table 1) of compound **1** showed signals for two methyl groups at δ_C_ 9.5 (C-1) and 14.2 (C-17); two O-bearing methine carbons at δ_C_ 64.1 (C-3) and 67.8 (C-11); two olefinic carbon resonances at δ_C_ 135.3 (C-9) and 124.4 (C-10); and seven methylene carbons with chemical shifts ranging from 18.3 to 37.4 ppm. The abovementioned signals accounted for one degree of unsaturation, and thus, the remaining required the existence of two additional triple bonds in the molecule, which consisted of four quaternary carbons δ_C_ 77.3 (C-4), 69.7 (C-5), 65.0 (C-6), and 78.6 (C-7) corresponding to the observed substitutions in the UV and IR spectra. By comparison, these data were relatively close to those of panaxjapyne A [6], except for the additional oxymethine at δ_C_ 67.8 (C-11), which was clearly proven by the ^1^H-^1^H COSY correlations (Figure 2) of H-11 with H-10 and H-12 and HMBC correlations (Figure 2) from H-11 to C-9, C-10, C-12, and C-13. Accordingly, compound **1** was elucidated as heptadeca-9-ene-4,6-diyne-3,11-diol.

Compound **2** was separated as a yellow oil with a negative optical rotation ${\left[\alpha \right]}_{D}^{25}$ −80 (*c* 0.1, CHCl_3_), and its molecular formula was determined to be C_17_H_26_O_3_ by the HRESIMS peak at *m*/*z* 301.1776 [M + Na]^+^ (calcd C_17_H_26_NaO_3_ for 301.1780). The ^1^H-NMR (Table 1) spectrum showed a pair of *trans* olefinic protons at δ_H_ 6.30 (1H, dd, *J* = 5.8, 15.9 Hz, H-9) and 5.86 (1H, d, *J* = 15.9 Hz, H-8); three oxymethine protons at δ_H_ 4.41 (1H, t, *J* = 6.5 Hz, H-3), 4.00 (1H, td, *J* = 5.8, 1.4 Hz, H-10), and 3.47 (1H, m, H-11); and two methyl groups at δ_H_ 1.01 (3H, t, *J* = 7.4 Hz, H-1) and 0.87 (3H, t, *J* = 6.9 Hz, H-17). A detailed analysis of the NMR spectroscopic spectra of compound **2** revealed that it is similar to panaxjapyne B [6] except for the additional oxymethine group in compound **2**. The ^1^H-^1^H COSY correlations (Figure 2) of H-8/H-9/H-10/H-11/H-12 and the HMBC correlations (Figure 2) from H-10 to C-11 and C-12 and from H-11 to C-9 and C-13 evidenced the location of oxymethine at C-11, which accounted for the molecular weight difference of 16 amu observed between the two compounds. Thus, the structure of compound **2** was elucidated as (*E*)-heptadeca-8-ene-4,6-diyne-3,10,11-triol. The relative configuration of C-10 and C-11 was determined by the ^3^*J*_H, H_ value and the ROESY interactions. The small ^3^*J*_H-10/H-11_ value (1.4 Hz) was indicative of a *gauche* relationship of H-10 and H-11 [34,35]. The ROESY correlations (Figure 3) of H-9/H-12/H-10/H-11 suggested that the relative configurations of C-10 and C-11 were *R** and *S**, respectively.

Compound **3** was separated as a yellow oil with a positive optical rotation ${\left[\alpha \right]}_{D}^{25}$ +80 (*c* 0.1, CHCl_3_). The molecular formula of compound **3** was established as C_18_H_30_O_3_ from the HRESIMS peak at *m*/*z* 317.2089 [M + Na]^+^ (calcd. C_18_H_30_NaO_3_ for 317.2093). The ^1^H and ^13^C-NMR spectroscopic data (Table 1) of compound **3** were comparable to those of oploxyne B [5], suggesting that an oxidized methine of oploxyne B was replaced by the methylene (δ_H_ 2.51, 2.57/δ_C_ 24.8) of compound **3**. Furthermore, the 16 amu of reduced molecular weight compared with oploxyne B indicated that the molecule formulas of the two compounds only differ by one oxygenium. The ^1^H-^1^H COSY correlations (Figure 2) of H-8/H-9/H-10/H-11 and the HMBC correlations (Figure 2) from H-8 to C-6, C-7, C-9, and C-10 implied that C-8 is the methylene carbon for compound **3**. Finally, the structure of compound **3** was elucidated as 10-methoxyheptadeca-4,6-diyne-3,9-diol. 

Compound **4** was obtained as a yellow oil with a negative optical rotation ${\left[\alpha \right]}_{D}^{25}$ −80 (*c* 0.1, CHCl_3_). The molecular formula was determined as C_14_H_22_O_3_ by the HRESIMS peak at *m/z* 261.1476 [M + Na]^+^ (calcd for C_14_H_22_NaO_3_, 261.1467). The ^1^H and ^13^C-NMR data (Table 2) showed three O-bearing methine groups at δ_H_ 4.13 (1H, br t, *J* = 6.5 Hz, H-6)/δ_C_ (69.6), 3.82 (1H, m, H-8)/δ_C_ (75.1), and 3.49 (1H, m, H-7)/δ_C_ (73.1); one triplet methyl group at δ_H_ 0.89 (3H, t, *J* = 6.7, H-14)/δ_C_ (14.2); one acetylene CH group at δ_H_ 2.00 (1H, s, H-1)/δ_C_ (65.6); three quaternary carbon groups at δ_C_ 68.2 (C-2), 67.0 (C-3), and 74.5 (C-4); and six methylenes at δ_H_ (1.30–2.64)/δ_C_ (22.7–33.6). The ^1^H and ^13^C-NMR spectroscopic data of compound **4** approached those of (6*S*,7*S*)-6,7-dihydroxytetradeca-1,3-diyne [36] except for an additional O-bearing methine in compound **4**. The structure was further elucidated by the HMBC and ^1^H-^1^H COSY data (Figure 2). The ^1^H-^1^H COSY correlations between H-5/H-6/H-7/H-8/H-9, together with the HMBC correlations from H-8 to C-6, C-9 (δ_C_ 33.6), and C-10 (δ_C_ 26.0) and from H-6 to C-4 and C-5 (δ_C_ 24.6) suggested that compound **4** has three successive O-bearing CH groups located at C-6, C-7, and C-8. Thus, the structure of compound **4** was elucidated as tetradeca-1,3-diyne-6,7,8-triol.

Compound **5** was obtained as a yellow oil with a negative optical rotation ${\left[\alpha \right]}_{D}^{25}$ −16 (*c* 0.1, CHCl_3_). The molecular formula was determined as C_22_H_30_O_8_ by the HRESIMS peak at *m*/*z* 445.1859 [M + Na]^+^ (calcd for C_22_H_30_NaO_8_, 445.1833). The ^1^H-NMR spectrum (Table 2) showed four O-bearing methine groups at δ_H_ 5.11 (1H, q, *J* = 5.6 Hz, H-6), 5.39 (1H, dd, *J* = 4.7, 6.2 Hz, H-7), 5.18 (1H, dd, *J* = 4.7, 6.2 Hz, H-8), and 5.07 (1H, q, *J* = 6.4 Hz, H-9); one triplet methyl group at δ_H_ 0.86 (3H, t, *J* = 6.9 Hz, H-14); one acetylene CH group at δ_H_ 2.00 (1H, s, H-1); 10 methylene protons at δ_H_ 1.67–2.62; and 12 acetoxy methyl protons at δ_H_ 2.10–2.12. The ^13^C-NMR and DEPT spectra indicated the presence of 22 carbons—one terminal methyl carbon at δ_C_ 14.1 (C-14); four O-bearing methine carbons at δ_C_ 69.1 (C-6), 70.6 (C-7), 71.2 (C-8), and 71.5 (C-9); three quaternary acetylenic carbons at δ_C_ 68.0 (C-2), 67.6 (C-3), and 71.6 (C-4); one tertiary acetylenic carbon at δ_C_ 66.0 (C-1); and five methylene carbons at 21.8–31.5 ppm. All the assignments were supported by the HSQC experiments. A detailed comparison of the spectroscopic data of compounds **4** and **5** showed that in compound **5**, there were four more acetyl groups [δ_C_ (170.0–170.6)/δ_H_ (2.10–2.12)] and one more O-bearing methine than in compound **4**. The ^1^H-^1^H COSY correlations (Figure 2) of H-5/H-6/H-7/H-8 and H-9/H-10/H-11, as well as the HMBC correlations (Figure 2) from H-6/H-6″ to C-6′ (170.1), from H-7/H-7″ to C-7′ (170.0), from H-8/H-8″ to C-8′ (170.3), and from H-9/H-9″ to C-9 (170.6), indicated that compound **5** has four acetyls linked to C-6, C-7, C-8, and C-9, respectively. The other key correlations of HMBC and ^1^H-^1^H COSY are shown in Figure 2. Thus, the planar structure of compound **5** was elucidated as 6,7,8,9-tetraacetoxytetradeca-1,3-diyne.

Due to the high flexibly of the carbon chains, crystals were not obtained from compounds **1**–**5**. Furthermore, the triple bonds contained compounds that were unstable and highly reactive [1], which resulted in all the isolates being chemically changed via oxidation and degradation. Therefore, the configurations of compounds **1**–**5** were not elucidated because of the materials’ instability and were reported as shown in Figure 1.

Compounds **1**–**7** were assessed for cytotoxic activity in the BEL-7402, K562, and SGC-7901 cancer cell lines, respectively. The results show that compounds **1** and **6** exhibited cytotoxicity in the above three human cancer lines, ranging from 14.3 to 45.4 μM (Table 3). Contrastingly, compound **4** displayed weak cytotoxicity in SGC-7901 and K562, with IC_50_ values of 26.2 ± 0.4 and 21.9 ± 0.3 μM, respectively. By comparison, the cytotoxicity in BEL-7401 and SGC-7901 was slightly enhanced when the double bond between C-9 and C-10 in compound **1** was oxidized to the epoxy group in compound **6**, which could be due to the highly genotoxic effect of the epoxy group [37,38].

## 3. Materials and Methods

### 3.1. General Experimental Procedures

The ^1^H, ^13^C, and 2D NMR spectra were recorded on a Bruker AV III spectrometer (Bruker, Bremen, Germany) at either 500 MHz (^1^H) or 125 MHz (^13^C) using TMS as an internal standard. The HRMS were measured with an API QSTAR Pulsar mass spectrometer (Bruker). The UV spectra were performed on a Shimadzu UV-2550 spectrometer (Beckman, Brea, CA, USA). The IR absorptions were obtained on a Nicolet 380 FT-IR instrument (Thermo, Pittsburgh, PA, USA) using KBr pellets. The optical rotation was measured on a Rudolph Autopol III polarimeter (Rudolph, Hackettstown, NJ, USA). Silica gel (60–80, 200–300 mesh, Qingdao Marine Chemical Co. Ltd., Qingdao, China), ODS gel (20–45 μm, Fuji Silysia Chemical Co. Ltd., Durham, NC, USA), and Sephadex LH-20 (Merck, Darmstadt, Germany) were used for column chromatography. The TLC was conducted on pre-coated silica gel G plates (Qingdao Marine Chemical Co. Ltd.), and spots were detected by spraying with 10% H_2_SO_4_ in EtOH followed by heating.

### 3.2. Plant Material

The plant material was collected at the Chinese Academy of Tropical Agricultural Sciences, Haikou, China, in April 2014 and was identified as *Swietenia macrophylla* by Dr. Jun Wang (Institute of Tropical Bioscience and Biotechnology, Chinese Academy of Tropical Agricultural Sciences). The voucher specimen (No. DYTHXM201404) was deposited at the Institute of Tropical Bioscience and Biotechnology, Chinese Academy of Tropical Agricultural Science.

### 3.3. Extraction and Isolation

The roots of *S. macrophylla* (57.0 kg, dry weight) were first crushed and extracted with 95% EtOH (3 × 150.0 L) at room temperature and evaporated to yield EtOH extract (8.0 kg), which was then partitioned with H_2_O (40.0 L) and extracted with petroleum ether (PE) (40.0 L × 3) and EtOAc (40.0 L × 3), respectively. The EtOAc extract (3931.0 g) was subjected to silica gel (20 × 50 cm, 12.0 kg) vacuum liquid chromatography, and eluted with CHCl_3_/MeOH (*v*/*v*, 10:1; 40.0 L) to obtain fraction 1 (Fr.1). Silica gel (10 × 50 cm, 2.0 kg) vacuum liquid chromatography of Fr.1 (573.0 g) was eluted with PE/EtOAc (*v*/*v*, 1:0, 100:1, 50:1, 25:1, 10:1, 5:1, each 5.0 L, gradient) and CHCl_3_/MeOH (*v*/*v*, 1:0, 100:1, 50:1, 25:1, 10:1, 5:1, 0:1, each 5.0 L, gradient), respectively, which resulted in 17 sub-fractions (Fr.1.1−Fr.1.17). Fr.1.8 (19.5 g) was applied to an ODS gel (4.5 × 40 cm) and eluted with MeOH/H_2_O (*v*/*v*, 3:7, 2:3, 1:1, 3:2, 7:3, 4:1, 9:1, 1:0, each 5.0 L) to yield Fr.1.8.1−10. Fr.1.8.5 (2.9 g) was separated on a Sephadex LH-20 (3 × 100 cm) with CHCl_3_/MeOH as the eluent (*v*/*v*, 1:1; 1.5 L) to give Fr.1.8.5.1−5. Fr.1.8.5.1 (1.6 g) was separated on a silica gel column (3 × 45 cm, 160 g) and eluted with PE/EtOAc (*v*/*v*, 12:1) to yield compound **1** (30.0 mg) before it was eluted with a gradient of PE/acetone (*v*/*v*, 20:1, 15:1, 5:1, 1:1, 0.6 L of each) to give six fractions (Fr.1.8.5.1.1−6). Compound **7** (2.2 mg) from Fr.1.8.5.1.1 (32.0 mg) was obtained from a silica gel (1 × 12 cm, 6 g) eluting with PE/EtOAc (*v*/*v*, 8:1). Compound **6** (30.0 mg) from Fr.1.8.5.1.2 (380.0 mg) was obtained from a silica gel (2 × 30 cm, 60.0 g) eluting with PE/CHCl_3_ (*v*/*v*, 1:1). Fr. 1.8.5.1.3 (150.0 mg) was subjected to a silica gel (1 × 25 cm, 15.0 g) with PE/EtOAc (*v*/*v*, 5:1) to obtain compound **3** (8.0 mg). Fr. 1.8.5.1.4 (180.6 mg) was applied to a silica gel (1 × 30 cm, 18.0 g) eluting with PE/EtOAc (*v*/*v*, 4:1) to yield compound **2** (14.8 mg). Fr. 1.8.5.1.5 (121.4 mg) was purified by a silica gel (1 × 30 cm, 20.0 g) eluting with CHCl_3_/acetone (*v*/*v*, 5:1) to obtain compounds **5** (1.2 mg) and **4** (3.8 mg).

*Heptadeca-9-ene-4,6-diyne-3,11-diol* (**1**): yellow oil; UV (CH_3_OH) *λ*_max_ (log *ε*) 280 (2.90), 270 (2.98) nm; IR (KBr) *ν*_max_ 3435, 2924, 2234, 1639, 1384, 1020 cm^−1^; ${\left[\alpha \right]}_{D}^{25}$ +10 (*c* 0.1, CH_3_OH); ^1^H and ^13^C-NMR data: Table 1; HRESIMS *m*/*z* 285.1830 [M + Na]^+^ (calcd. C_17_H_26_NaO_2_ for 285.1831).

*(E)-Heptadeca-8-ene-4,6-diyne-3,10,11-t**riol* (**2**): yellow oil; UV (CHCl_3_) *λ*_max_ (log *ε*) 286 (3.60), 270 (3.68) nm; IR (KBr) *ν*_max_ 3433, 2927, 2250, 1645, 1382, 1026 cm^−1^; ${\left[\alpha \right]}_{D}^{25}$ −80 (*c* 0.1, CHCl_3_); ^1^H and ^13^C-NMR data: Table 1; HRESIMS *m*/*z* 301.1776 [M + Na]^+^ (calcd. C_17_H_26_NaO_3_ for 301.1780).

*10-Methoxyheptadeca-4,6-diyne-3,9-diol* (**3**): yellow oil; UV (CHCl_3_) *λ*_max_ (log *ε*) 286 (2.77), 270 (2.78) nm; IR (KBr) *ν*_max_ 3432, 2929, 2245, 1388, 1028 cm^−1^; ${\left[\alpha \right]}_{D}^{25}$ +80 (*c* 0.1, CHCl_3_); ^1^H and ^13^C-NMR data: Table 1; HRESIMS *m*/*z* 317.2089 [M + Na]^+^ (calcd. C_18_H_30_NaO_3_ for 317.2093).

*Tetradeca-1,3-diyne-6,7,8-triol* (**4**): yellow oil; UV (CHCl_3_) *λ*_max_ (log *ε*) 298 (1.32), 240 (2.60) nm; IR (KBr) *ν*_max_ 3434, 2926, 2241, 1384, 1018 cm^−1^; ${\left[\alpha \right]}_{D}^{25}$ −80 (*c* 0.1, CHCl_3_); ^1^H and ^13^C-NMR data: Table 2; HREIMS *m*/*z* 261.1476 [M + Na]^+^ (calcd for C_14_H_22_NaO_3_, 261.1467). 

*6,7,8,9-Tetraacetoxytetradeca-1,3-diyne* (**5**): yellow oil; UV (CHCl_3_) *λ*_max_ (log *ε*) 212 (4.44), 271 (3.55), 308 (3.12) nm; IR (KBr) *ν*_max_ 2924, 2233, 1741, 1634, 1460, 1090 cm^−1^; ${\left[\alpha \right]}_{D}^{25}$ −16 (*c* 0.1, CHCl_3_); ^1^H and ^13^C-NMR data: Table 2; HREIMS *m*/*z* 445.1859 [M + Na]^+^ (calcd for C_22_H_30_NaO_8_, 445.1833). 

### 3.4. Bioassay of Cytotoxic Activity

MTT assay, originally described by Mosmann [39], was used to quantitate the cytotoxicity of compounds **1**–**7**. The human hepatocellular carcinoma cell line BEL-7402, human myeloid leukemia cell line K562, and human gastric carcinoma cell line SGC-7901, which were obtained from the cell bank of type culture collection of the Chinese Academy of Sciences, Shanghai Institute of Cell Biology, were cultured in RPMI 1640 medium supplemented with 10% fetal bovine serum at the conditions of 37 °C, 5% CO_2_, and 90% humidity. Paclitaxel was used as the positive control and DMSO was used as the negative control. Different concentrations of the test sample (each had triplicate wells) were designed as 0.1, 0.4, 1.6, 6.3, 25, and 100 μM. The logarithmic phase cells (90 μL) were selected to seed onto the 96-well plates at a concentration of 5 × 10^4^ cell/mL. Then, 15 μL of MTT dissolved in PBS at 5 mg/mL was added to each well, and the system was incubated at 37 °C for 4 h. After that, the supernatant was discarded, and 100 μL of DMSO was added into each well. Finally, the OD value was measured by a MK3 Microtiter plate reader at a wavelength of 490 nm.

## 4. Conclusions

Seven polyacetylenes were isolated from the roots of *S. macrophylla*. Their structures were determined by spectroscopic analysis and comparing data in the literature. Furthermore, compounds **1** and **6** displayed weak cytotoxicity in the BEL-7402, SGC-7901, and K562 cell lines, and compound **4** showed a weak cytotoxic effect in the SGC-7901 and K562 cell lines.

## Figures and Tables

**Figure 1 molecules-24-01291-f001:**
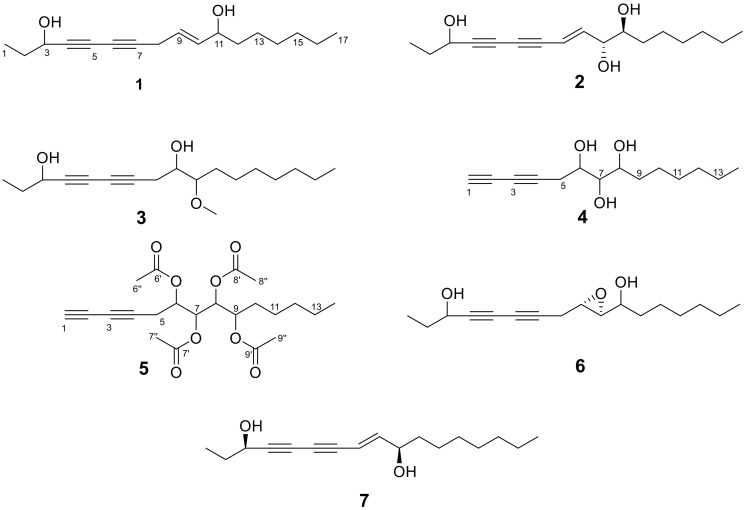
Structures of compounds **1**–**7**.

**Figure 2 molecules-24-01291-f002:**
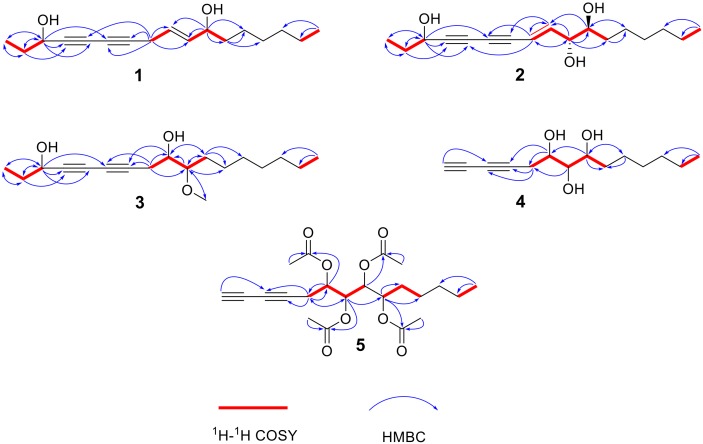
Key ^1^H-^1^H COSY and HMBC correlations of compounds **1**–**5**.

**Figure 3 molecules-24-01291-f003:**
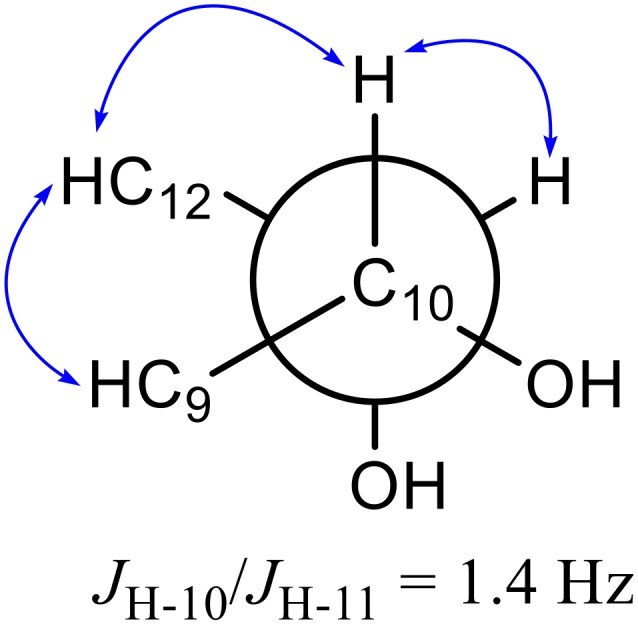
Key ROESY interactions of compound **2**.

**Table 1 molecules-24-01291-t001:** ^1^H (500 MHz) and ^13^C-NMR (125 MHz) data for compounds **1**–**3** in CDCl_3_ (δ in ppm, *J* in Hz).

Position	1		2		3	
	δ_H_	δ_C_	δ_H_	δ_C_	δ_H_	δ_C_
1	0.99, t (7.4)	9.5, CH_3_	1.01, t (7.4)	9.5, CH_3_	1.01, t (7.4)	9.5, CH_3_
2	1.72, m	30.8, CH_2_	1.75, m	30.8, CH_2_	1.73, m	30.8, CH_2_
3	4.34, t (6.4)	64.1, CH	4.41, t (6.5)	64.3, CH	4.35, t (6.4)	64.2, CH
4		77.3, C		83.5, C		77.0, C
5		69.7, C		69.5, C		69.8, C
6		65.0, C		74.4, C		66.3, C
7		78.6, C		76.7, C		77.8, C
8	3.09, dd (5.0, 10.2)	18.3, CH_2_	5.86, d (15.9)	110.5, CH	2.51, dd (17.3, 6.1) 2.57, dd (17.3, 6.3)	24.8, CH_2_
9	5.50, overlapped	135.3, CH	6.30, dd (5.8, 15.9)	146.3, CH	3.70, td (6.3, 4.3)	71.0, CH
10	5.48, overlapped	124.4, CH	4.00, td (5.8, 1.4)	75.2, CH	3.24, td (6.1, 4.3)	82.1, CH
11	4.39 q (6.7)	67.8, CH	3.47, m	74.4, CH	1.55, m	29.9, CH_2_
12	1.60, m; 1.44, m	37.4, CH_2_	1.46, m	33.2, CH_2_	1.28–1.29, m	25.3, CH_2_
13	1.27–1.28, m	25.3, CH_2_	1.27–1.28, m	25.7, CH_2_	1.28–1.29, m	29.4, CH_2_
14	1.27–1.28, m	29.3, CH_2_	1.27–1.28, m	29.4, CH_2_	1.28–1.29, m	29.9, CH_2_
15	1.27–1.28, m	31.9, CH_2_	1.27–1.28, m	31.9, CH_2_	1.28–1.29, m	31.9, CH_2_
16	1.27–1.28, m	22.7, CH_2_	1.27–1.28, m	22.7, CH_2_	1.28–1.29, m	22.8, CH_2_
17	0.87 t (6.8)	14.2, CH_3_	0.87 t (6.9)	14.2, CH_3_	0.88 t (6.8)	14.2, CH_3_
−OCH_3_					3.42, s	58.5, CH_3_

**Table 2 molecules-24-01291-t002:** ^1^H (500 MHz) and ^13^C-NMR (125 MHz) data for compounds **4** and **5** in CDCl_3_ (δ in ppm, *J* in Hz).

Position	4		5				
	δ_H_	δ_C_	δ_H_	δ_C_	Position	δ_H_	δ_C_
1	2.00, s	65.6, CH	2.00, s	66.0, CH	6′		170.1, C
2		68.2, C		68.0, C	7′		170.0, C
3		67.0, C		67.6, C	8′		170.3, C
4		74.5, C		71.6, C	9′		170.6, C
5	2.58, dd (17.4, 6.8) 2.64, dd (17.4, 6.5)	24.6, CH_2_	2.62, d (5.6)	21.8, CH_2_	6″	2.11 ^b^, s	20.8 ^a^, CH_3_
6	4.13, br t (6.5)	69.6, CH	5.11, q (5.6)	69.1, CH	7″	2.12 ^b^, s	21.0 ^a^, CH_3_
7	3.49, m	73.1, CH	5.39, dd (4.7, 6.2)	70.6, CH	8″	2.10 ^b^, s	20.9 ^a^, CH_3_
8	3.82, m	75.1, CH	5.18, dd (4.7, 6.2)	71.2, CH	9″	2.10 ^b^, s	20.8 ^a^, CH_3_
9	1.55, m	33.6, CH_2_	5.07, q (6.4)	71.5, CH			
10	1.33, m1.55, m	26.0, CH_2_	1.55, m	30.6, CH_2_			
11	1.30, br s	29.4, CH_2_	1.28, m	24.6, CH_2_			
12	1.30, br s	31.9, CH_2_	1.28, m	31.5, CH_2_			
13	1.30, br s	22.7, CH_2_	1.27, m	22.5, CH_2_			
14	0.89, t (6.7)	14.2, CH_3_	0.86, t (6.9)	14.1, CH_3_			

^a,b^ exchangeable.

**Table 3 molecules-24-01291-t003:** Cytotoxicity of compounds **1**–**7** in the human cancer cell lines.

Compound	Cell Line, IC_50_ (μM)		
	BEL-7402	SGC-7901	K562
**1**	24.9 ± 0.3	45.4 ± 0.6	16.8 ± 0.1
**4**	>50	26.2 ± 0.4	21.9 ± 0.3
**6**	14.3 ± 0.4	33.4 ± 0.6	16.6 ± 0.4
**2**, **3**, **5**, **7**	>50	>50	>50
Paclitaxel ^a^	4.3 ± 0.1	4.3 ± 0.2	8.6 ± 0.1

^a^ Positive control.

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
