# Peer review of "Polyacetylenes from the Roots of Swietenia macrophylla King"

_molecules, 2019, doi:10.3390/molecules24071291_

Round 1
Reviewer 1 Report
The manuscript in reference describes the isolation of five novel polyacetylenes along two known polyacetylenes and their cytotoxic activity against three cell lines. In general, the manuscript is well-written and has important finding that deserve publication in molecules. In addition, the structural elucidation is well-conducted and finely described. However, some minor points must be addressed prior acceptance.
Revise the grammar since some errors can be found in several parts of the manuscript. A detailed and carefully revision is therefore required.
Page 1, line 27. The name of the family must be Asteraceae instead Compositae.
Page 4, line 94. Revise the name of compound 2.
Page 4, line 121. Revise the name of compound 4 and unify the name format with other novel compounds 1-3.
Introduction: It is not clear why authors evaluated cytotoxic activity for the isolated compounds, since only it is indicated the aim of the work was focused on extend the biological activity of the plant source, but this indication is very ambiguous. Any previous study? Ethnobotany? a particular research line? I recommend to include a justification about that in order to clarify.
In the text, explicit the carbon number for compound 3 to be different from oploxyne B, because it is not clear and the reader must deduce that.
Conclusions section is missing.
According to the above-mentioned points, I recommend acceptance after minor revisions.
Author Response
Dear Editor,
Thank you for your letter and for the reviewers’ comments concerning our manuscript entitled “Polyacetylenes from the roots of Swietenia macrophylla King”. Those comments are all valuable and helpful for revising and improving our paper, as well as the important guiding significance to our researches. We have studied comments carefully and have made correction which we hope meet with approval. Revised portion are marked in red in the paper. The main corrections in the paper and the responds to the reviewer’s comments are as flowing:
The manuscript in reference describes the isolation of five novel polyacetylenes along two known polyacetylenes and their cytotoxic activity against three cell lines. In general, the manuscript is well-written and has important finding that deserve publication in molecules. In addition, the structural elucidation is well-conducted and finely described. However, some minor points must be addressed prior acceptance.
Response:Thank you very much for your admiring about our works and we also sorry about our careless on manuscript. We revised some problems and the insufficiency from your suggested.
Point 1: Page 1, line 27. The name of the family must be Asteraceae instead Compositae.
Response 1: The word “Compositae” we used in the article and highlights were replaced by “Asteraceae”
Point 2: Page 4, line 94. Revise the name of compound 2.
Response 2: Compound 2 was revised to “(E)-Heptadeca-8-ene-4,6-diyne-3,10,11-tirol”.
Point 3: Page 4, line 121. Revise the name of compound 4 and unify the name format with other novel compounds 1-3
Response 3: Compound 4 was revised to “tetradeca-1,3-diyne-6,7,8-triol”.
Point 4: Introduction: It is not clear why authors evaluated cytotoxic activity for the isolated compounds, since only it is indicated the aim of the work was focused on extend the biological activity of the plant source, but this indication is very ambiguous. Any previous study? Ethnobotany? a particular research line? I recommend to include a justification about that in order to clarify.
Response 4: So far, the published research on Swietenia macrophylla has been focused mostly on aboveground parts and their limonoid components; but other plant parts and other compound types may also have rich pharmacological activities. Therefore, it is extremely urgent to expand the scope of research on Swietenia macrophylla and discover or develop additional biologically active constituents of this plant [Molecules 2018, 23(7): 1588]. Cytotoxicity is important featuring activity of some polyacetylenes, as well as compound 6 have exhibited potent cytotoxicity against the HL-60 cell line with an IC50 value of 6.7 ± 0.27 mm in references 18 [Helv. Chim. Acta. 2011, 94(3): 376—381], so we evaluated cytotoxic activity, at first.
Point 5: In the text, explicit the carbon number for compound 3 to be different from oploxyne B, because it is not clear and the reader must deduce that.
Response 5: We have revised the structure analysis in the manuscript, the reader can see clearly.
Point 6: Conclusions section is missing.
Response 6: Conclusions were appended in the manuscript.
Reviewer 2 Report
The manuscript presents a careful study on extraction, separation, identification and cytotoxicity of polyacetylenes from the roots of Swietenia macrophylla King. Due to the identification of five new naturally occouring polyacetylenes, the manuscript is of significant interest to the readers of Molecules. However, the following issues have to be successfully resolved before its publication could be recommended:
1) Investigation of bilogical activity of newly identified polyacetylenes is very limited including only the MTT test of cytotoxicity. In accordance with references [Nutrients 2019, 11, 257] this reviewer would expect the bioactivity studies to be extended to determination of their antioxidative (using DPPH or ABTS tests) and atimicrobial potentials.
2) The enhanced citotoxicity of compound 6 in comparison with compound 1 should be of no surprise, because the epoxy group has been shown to be highly genotoxic by both experimental [Chem. Res. Toxicol. 2015, 28: 691-701] and computational [Molecules 2019, 24, 150] studies. Quote and discuss.
3) Page 6, line 145: Therefore, the configurations of 1-5 WERE not elucidated because of the material instability...
Author Response
Dear Editor,
Thank you for your letter and for the reviewers’ comments concerning our manuscript entitled “Polyacetylenes from the roots of Swietenia macrophylla King”. Those comments are all valuable and helpful for revising and improving our paper, as well as the important guiding significance to our researches. We have studied comments carefully and have made correction which we hope meet with approval. Revised portion are marked in red in the paper. The main corrections in the paper and the responds to the reviewer’s comments are as flowing:
The manuscript presents a careful study on extraction, separation, identification and cytotoxicity of polyacetylenes from the roots of Swietenia macrophylla King. Due to the identification of five new naturally occouring polyacetylenes, the manuscript is of significant interest to the readers of Molecules. However, the following issues have to be successfully resolved before its publication could be recommended:
Thank you very much for your admiring about our works and we also sorry about our careless on manuscript. We revised some problems and the insufficiency from your suggested.
Point 1: Investigation of bilogical activity of newly identified polyacetylenes is very limited including only the MTT test of cytotoxicity. In accordance with references [Nutrients 2019, 11, 257] this reviewer would expect the bioactivity studies to be extended to determination of their antioxidative (using DPPH or ABTS tests) and atimicrobial potentials.
Response 1: Compound 6 had exhibited potent cytotoxicity against the HL-60 cell line with an IC50 value of 6.7 ± 0.27 μM in references 18 [Helv. Chim. Acta. 2011, 94(3): 376−381], so we evaluated cytotoxic activity, at first. However, the consuming of material for testing cytotoxicity makes us unable to test other bioassay, especially for those compounds which we isolated only with little amount. We have carefully considered that evaluated antioxidative and antimicrobial potentials of compound 1−3 and 6, but those compounds have been verified to be change by both 1H NMR, HPLC and TLC analysis. So, we feel very regret for not collected the data onto the antioxidative and atimicrobial activity.
Point 2: The enhanced citotoxicity of compound 6 in comparison with compound 1 should be of no surprise, because the epoxy group has been shown to be highly genotoxic by both experimental [Chem. Res. Toxicol. 2015, 28: 691-701] and computational [Molecules 2019, 24, 150] studies. Quote and discuss.
Response 2: Thank you very much for your suggestion about our article and references. After carefully and completely read of the supplied references, we can benefit a lot from references. So, we had been quoted and discussed those references.
Point 3: Page 6, line 145: Therefore, the configurations of 1-5 WERE not elucidated because of the material instability....
Response 3: We revised this inappropriate part in the manuscript.

Round 2
Reviewer 2 Report
The authors successfully resolved two issues raised by this reviewer. The third issue could not be resolved due to the lack of the isolated material. All in all, the manuscript has been significantly improved from its previous version and could be in its current form recommended for the publication in Molecules.
Author Response
Thanks for your attention to this article.